# The Effect of Nitrogen and Potassium Interaction on the Leaf Physiological Characteristics, Yield, and Quality of Sweet Potato

Xing Shu [†], Minghuan Jin [†], Siyu Wang, Ximing Xu , Lijuan Deng, Zhi Zhang, Xu Zhao, Jing Yu, Yueming Zhu, Guoquan Lu  and Zunfu Lv *

The Key Laboratory for Quality Improvement of Agricultural Products of Zhejiang Province, College of Advanced Agricultural Sciences, Zhejiang A&F University, Lin'an District, Hangzhou 311300, China; shuxing@stu.zafu.edu.cn (X.S.); 2021101011014@stu.zafu.edu.cn (M.J.); wangsiyu@stu.zafu.edu.cn (S.W.); xuximing@zafu.edu.cn (X.X.); 2020601042004@stu.zafu.edu.cn (L.D.); zhangzhi0908@stu.zafu.edu.cn (Z.Z.); 2023101011026@stu.zafu.edu.cn (X.Z.); 2024601022037@stu.zafu.edu.cn (J.Y.); zhuym@zafu.edu.cn (Y.Z.); lugq@zafu.edu.cn (G.L.)

* Correspondence: 20140001@zafu.edu.cn
† The first two authors contributed equally to this work.

**Abstract:** This study selected two sweet potato varieties as research subjects and conducted a field experiment using a two-factor design with two potassium (K) levels (K0 and K1) and five nitrogen (N) levels (N0–N4). The physiological changes in sweet potato leaves under different N and K treatments were measured, and nutrients such as the soluble sugar, protein, and starch content of sweet potato roots were analyzed. The results indicate that the activity of glutamine synthetase (GS) and the soluble protein content in sweet potato leaves increase first and then decrease with increasing N application, while K application can significantly increase the activity of GS and the soluble protein content. The N metabolic capacity of leaves is strongest when the fertilizer ratio is K1N2. The SPAD value of sweet potato leaves increases with increasing N application. The net photosynthetic rate, stomatal conductance, and intercellular $CO_2$ concentration first increase and then decrease with increasing N application. K fertilizer has a significant effect on these parameters. As the N application rate increases, the starch and protein content in the tubers increase, while the soluble sugar content decreases. However, the number of tubers per plant, fresh weight of the tubers, and dry weight of the tubers increase initially and then decrease, while the vine length continuously increases. The application of K fertilizer can significantly increase the number of tubers per plant and stem thickness of sweet potato. In conclusion, the appropriate N–K combined application can promote N metabolism, enhance the photosynthetic capacity of sweet potato, increase yield, and improve quality.

**Keywords:** nitrogen–potassium interaction; photosynthesis; glutamine synthetase; tuber yield; tuber quality; sweet potato

## 1. Introduction

Sweet potato is one of the important industrial crops in the world and an important raw material for food processing, and the starch and alcohol manufacturing industries [1,2]. In the process of plant physiological metabolism, nitrogen (N) and potassium (K) in plants have complementary effects. Unreasonable N and K ratios will have a significant impact on the growth and development of crops, and will have adverse effects on the stress resistance, yield, and quality of crops [3,4].

N metabolism plays a crucial role in the life processes of plants, providing a fundamental material basis for their growth and development [5]. Within the process of N assimilation, glutamine synthetase (GS) is a key rate-limiting enzyme in plant N metabolism, converting inorganic N absorbed by plants into amino acids to be absorbed and utilized by plants [6]. By appropriately increasing the amount of N fertilizer, the efficiency of N assimilation

can be improved and protein synthesis can be promoted [7,8]. The level of the soluble protein content of plant leaves can reflect the level of plant metabolism [9]. Studies have shown that, under different N application levels, the soluble protein content in sugar beet leaves is positively correlated with GS activity in early-stage leaves [10]. Photosynthesis is a core physiological function in plants, which can be used to characterize the health and vitality of plants [11]. The photosynthesis of plants is, to some extent, affected by N levels. N deficiency leads to a significant decrease in the photosynthetic capacity of plants, while an increase in leaf N content helps to improve the conduction of $CO_2$ in leaves, and increase the leaf stomatal conductance and mesophyll conductance, thereby enhancing the leaf photosynthetic capacity [12]. K primarily functions in plants by maintaining cellular osmotic balance, enhancing the activity of key photosynthetic enzymes, promoting stomatal movement and the transport of photosynthetic products, as well as accelerating $CO_2$ fixation, thereby improving the plant photosynthetic capacity [13,14]. Sweet potato exhibits a typical source–sink relationship. N can improve the photosynthetic performance of sweet potato, accelerate the conversion of photosynthetic products, and promote nutrient accumulation. Additionally, increased K fertilizer application plays an important role in increasing the chlorophyll content in sweet potato leaves, promoting the transport of carbohydrates from leaves to tubers, and facilitating tuber enlargement [15,16].

As a typical tuberous root crop, sweet potato mainly consists of tuberous roots; therefore, obtaining the maximum dry matter mass underground with the most suitable fertilizer ration is the ultimate goal of agricultural cultivation. The formation and expansion of sweet potato tuberous roots is a complex process [17,18]. Tuberous roots develop from adventitious roots, with the formation layer cells in adventitious roots undergoing division, producing a large number of thin-walled cells that continuously accumulate starch, thereby developing into tuberous roots [19]. Starch is the main tuberous form in sweet potato and is an important factor determining the sweet potato yield, while the soluble sugar content in different parts of sweet potato has a direct relationship with the rate of starch accumulation [20]. N can increase the photosynthetic capacity of sweet potato, enhance photosynthetic product accumulation, and promote nutrient accumulation. However, excessive N fertilizer can lead to the excessive growth of stems and leaves above ground, which is not conducive to the enlargement of tuberous roots, resulting in reduced yield [21]. K can enhance the activity of leaf sucrose synthase and promote the transport of photosynthetic products to the underground parts, providing an adequate material basis for starch synthesis, thereby promoting root enlargement and increasing the sweet potato yield [22]. The nutritional interaction of N and K jointly affects the growth and development of crops, thereby affecting the crop yield and quality. The combined application of N and K fertilizers promotes rice leaf development, increases light energy utilization, and significantly increases the rice yield and yield components [23]. Research has found that an appropriate N–K ratio can significantly increase the protein and soluble sugar content of sweet potato tubers and improve the quality of sweet potato [24]. The rational combination of N and K can significantly increase the number of tubers per plant and the weight of individual tubers of sweet potato, and increase the expansion rate of sweet potato roots and the rate of commercial tubers, thereby increasing the yield of sweet potato tubers [25].

The effects of N and K combined fertilization on the growth, development, yield, and quality of rice, wheat, and other crops have been widely reported. However, there are few studies on the effects of N and K combined fertilization on the N metabolism, photosynthetic characteristics, yield, and quality of sweet potato during the root expansion period. Therefore, this study aimed (1) to investigate the effect of the N and K ratio on sweet potato N metabolism, photosynthetic characteristics, yield, and quality, so as to clarify the physiological mechanism of the N and K interaction; and (2) to clarify the impact of aboveground photosynthetic physiology characteristics on underground yield and quality.

## 2. Materials and Methods

### 2.1. Experimental Site Overview

This experiment was conducted in Guantang, Zhejiang A&F University (30°23′ N, 119°73′ E) during 2021. The area comes under subtropical monsoon climate, in a warm and humid environment with sufficient light, with abundant rainfall during rainy season. The average annual precipitation is 1613.9 mm, with 158 rainy days and an average annual frost-free period of 237 days. Soil samples were collected and analyzed from the 0–20 cm soil tillage layer before planting. It was determined that the basic fertility N content of the soil was 1.34 g·kg$^{-1}$, the available phosphorus content was 108.9 mg·kg$^{-1}$, and the available K content was 116 mg·kg$^{-1}$.

### 2.2. Experimental Design

In this study, starchy sweet potato 'Shang19' and fresh sweet potato 'Yan25' were selected as experimental materials. Both varieties have been widely promoted, which have earned high market recognition and high economic value. The field experiment adopted a split-plot design, with K fertilizer in the main plot and N fertilizer in the secondary plot. The experiment set five N fertilizer application levels (N0: 0 kg·ha$^{-1}$, N1: 60 kg·ha$^{-1}$, N2: 120 kg·ha$^{-1}$, N3: 180 kg·ha$^{-1}$, and N4: 240 kg·ha$^{-1}$) and two K fertilizer (K$_2$O) application levels (K0: 0 kg·ha$^{-1}$ and K1: 240 kg·ha$^{-1}$). Each treatment was repeated three times, and a total of 20 plots were set. In addition, 60 kg·ha$^{-1}$ of P$_2$O$_5$ was applied as basal fertilizer in combination with land preparation. N fertilizer was applied once as basal fertilizer, and no topdressing was used in the experiments. The crop was planted by vine cuttings with the spacing of 25 cm × 80 cm under the black plastic mulch.

### 2.3. Measurement Content and Method

#### 2.3.1. Metabolic Enzyme Assay

The soluble protein content was measured using the Coomassie Brilliant Blue G-250 colorimetric method [26]; the GS activity was measured using the FeCl$_3$ colorimetric method [27]. First, the enzyme solution was extracted from 1.0 g sweet potato leaves with 3 mL extraction buffer. Next, 0.7 mL of the enzyme solution was combined with 1.6 mL of the reaction mixture and 0.7 mL of ATP solution, mixed thoroughly, and insulated at 37 °C for 30 min. Then, 1.0 mL of color developer was added, mixed well, and centrifuged. The supernatant was collected, and its absorbance was measured at 540 nm. Additionally, 0.5 mL of the extracted enzyme solution was diluted to 100 mL with distilled water. Then, 2 mL of the diluted solution was used to measure the soluble protein content at 595 nm using Coomassie Brilliant Blue G-250.

#### 2.3.2. Determination of Photosynthetic Characteristics

At about 80 days after planting, SPAD values were measured with SPAD-502 chlorophyll meter in the morning on a sunny day, three uniformly growing plants were taken for each treatment, and seven to eight nodes of functional leaves of each plant were measured, and a total of six pieces of data were measured for each treatment, the average of which was the SPAD value of the treatment; functional leaves of sweet potato with robust and uniform growth were selected, and the net photosynthetic rate (Pn), transpiration rate (Tr), intercellular CO$_2$ concentration (Ci), and stomatal conductance (Gs) of sweet potato leaves were measured using Li-6400 photosynthetic instrument from 9:00 a.m. to 11:00 a.m. Three leaves were taken for each treatment, and each leaf was measured three times, with a total of nine pieces of data measured for each treatment, and a mean value was taken.

#### 2.3.3. Root Quality and Yield Determination

Quality measurement: About 80 days after planting, we took 3 evenly grown plants from each treatment. We chose about 1 kg of moderately sized fresh sweet potato pieces, washed, dried, and chopped them, and then weighed about 200 g using the quartering method. As a sample, the sample was placed in an oven at 105 °C. After curing for 30 min,

it was dried at 60 °C. The sample was crushed using a cyclone mill for the determination of quality indicators. Firstly, 0.5 g of sample powder was weighed and added to 80% ethanol. The mixture was extracted 2–3 times at 80 °C, and the supernatant obtained was used to measure soluble sugars. The remaining residue was then extracted with distilled water and $HClO_4$ in a boiling water bath, and the supernatant was used as a starch test solution. A 2 mL test solution was mixed with anthrone–$H_2SO_4$ reagent and heated in a boiling water bath for 10 min. After cooling, the absorbance was measured at 620 nm [28]; in addition, 0.2 g of powder sample was placed in a digestion tube. Next, 10 mL of concentrated $H_2SO_4$ was added to digest at 420 °C for 1.5 h, and then $H_2O_2$ was added for reaction. After cooling, the volume was 50 mL with distilled water. The solution was subsequently processed using a Kjeldahl N autoanalyzer, distilled, and titrated, and the titration data were recorded to calculate the protein content [26].

Yield measurement: During the harvest period (130 days after planting), 3 plants from each treatment were randomly selected, and the number of tubers per plant, fresh weight, vine length, and stem diameter were counted, respectively. We selected about 1 kg of moderately sized fresh sweet potato pieces, washed, dried, and chopped them, and then weighed about 200 g using the quartering method. Then, we used the quartering method to weigh about 200 g as a sample. The samples were placed in an oven at 105 °C, dried for 30 min, and then dried at 60 °C and weighed.

### 2.4. Statistical Analysis

Microsoft Office Excel 2016 was used for data entry, organization, and preliminary calculation. The data were analyzed by two-way ANOVA and Duncan's multiple range test ($p < 0.05$) using SPSS 22.0 to evaluate the effects of N and K treatments and their interactions on various physiological parameters of sweet potato plants. GraphPad Prism 9.5 was used for presentation of figures.

## 3. Results

### 3.1. Leaf Metabolic Enzyme Activity Analysis

3.1.1. Leaf Soluble Protein Analysis

With the increasing amount of N application, the soluble protein content in sweet potato leaves showed a tendency to increase first and then decrease (Figure 1). The application of a certain amount of N fertilizer was able to significantly increase the soluble protein content in sweet potato leaves ($p < 0.05$).

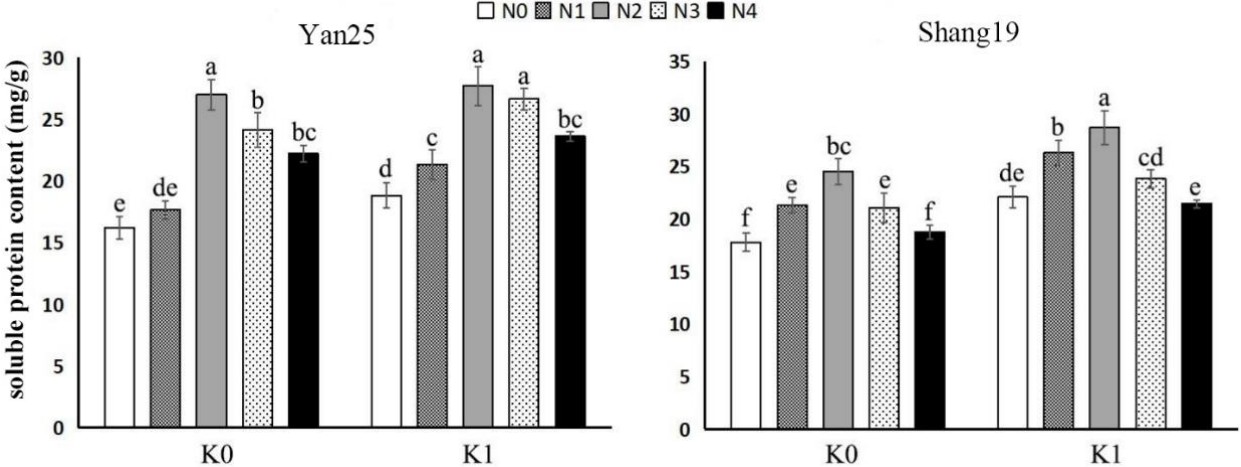

**Figure 1.** Changes in soluble protein content in sweet potato leaves under different N and K levels. The letters on the column represent the significant difference at the 5% level. Columns with the same letter have no significant difference at $p < 0.05$.

The soluble protein content increased continuously from N0 to N2 treatments and reached the maximum value at the N2 treatment. The soluble protein content of Yan25 was increased by 66.7% and 47.1% at the N2 treatment compared with the N0 treatment at the K0 and K1 levels, respectively; and the soluble protein of Shang19 was increased by 3% and 9% at the K0 and K1 levels compared with the N0 treatment by 3% and 9%, respectively. Compared with the K0 treatment, K application increased the soluble protein content of the leaves of Yan25 by 2.7–20.8% and that of Shang19 by 13.3–24.2%.

The two-factor analysis showed that the application of K fertilizer significantly increased the soluble protein content in leaves, while the interaction between the N and K fertilizer on the soluble protein content in leaves was not significant (Table 1). In this study, the soluble protein content of Yan25 and Shang19 had the highest soluble protein content when the ratio of N and K was K1N2.

**Table 1.** Interaction analysis of N and K combined application on soluble protein and GS enzyme.

| Varieties | ANOVA | F Value | |
| --- | --- | --- | --- |
| | | Soluble Protein | GS Enzyme |
| | N | 63.1 *** | 75.1 *** |
| Yan25 | K | 23.0 ** | 302.0 *** |
| | N × K | 1.3 ns | 24.2 *** |
| | N | 46.9 *** | 30.3 *** |
| Shang19 | K | 109.4 *** | 61.0 *** |
| | N × K | 1.5 ns | 10.9 ** |

** $p < 0.01$, *** $p < 0.001$.

### 3.1.2. Leaf GS Analysis

With the continuous increase in N application, the GS activity of Yan25 and Shang19 leaves showed a tendency of first increasing and then decreasing. For N application from N0 to N2, the GS activity increased continuously, and reached a maximum in the N2 treatment (Figure 2).

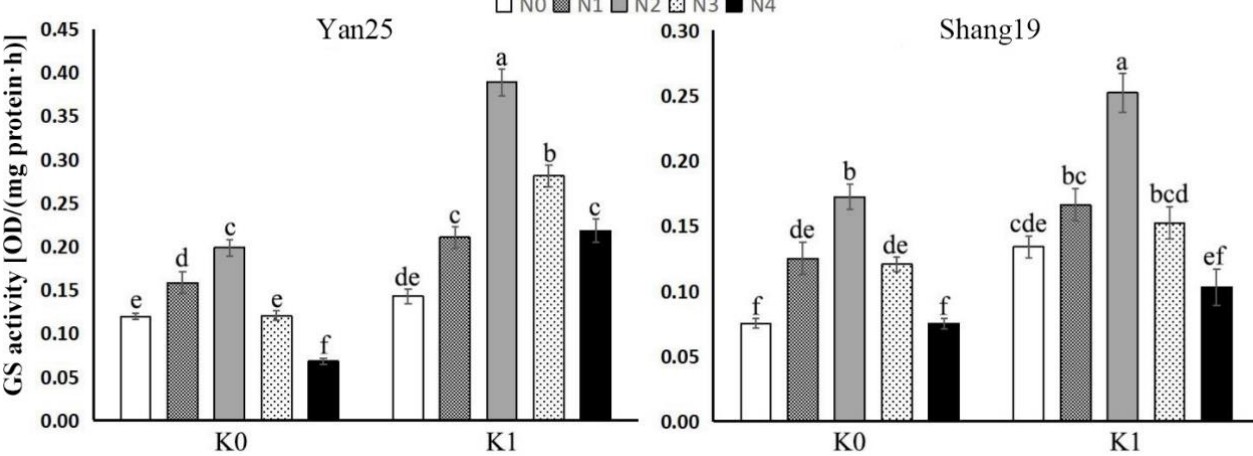

**Figure 2.** Changes in GS activity in sweet potato leaves under different N and K levels. The letters on the column represent the significant difference at the 5% level. Columns with the same letter have no significant difference at $p < 0.05$.

The application of a certain amount of N fertilizer was able to significantly increase the GS activity in sweet potato leaves ($p < 0.05$). The GS activity of Yan25 was increased by 0.66% and 1.72% in the N2 treatment compared with the N0 treatment at the K0 and K1 levels, respectively; and the GS activity of Shang19 was increased by 1.28% and 0.88% in the N2 treatment compared with the N0 treatment at the K0 and K1 levels, respectively.

Compared with the K0 treatment, the GS activity in the leaves of Yan25 was increased by 19.6–221.7% and that of Shang19 was increased by 26.4–78% after K application.

The two-factor analysis showed that the application of K fertilizer significantly increased the GS activity in leaves, and there was a significant interaction effect of the N and K fertilizer dosing on the GS activity in leaves (Table 1). The highest GS activity in the leaves of Yan25 and Shang19 in this study was found when the fertilizer ratio was K1N2.

### 3.2. Leaf Photosynthetic Characteristics Analysis

### 3.2.1. SPAD Value

From Figure 3, it can be seen that the SPAD values of the leaves of Yan25 and Shang19 increased continuously with the increasing amount of N applied. Applying the N fertilizer can significantly increase the SPAD value of sweet potato leaves ($p < 0.05$). Under the K0 and K1 levels, when the N fertilizer was applied, the SPAD value of the leaves of Yan25 increased by 2.3–19.3% and 2.8–24.7%, respectively, compared with the N0 treatment. The SPAD value of the leaves of Shang19 increased compared with the N0 treatment, 0–16.5% and 1.6–16.7%. The K fertilizer had no significant effect on the leaf SPAD value. The two-factor analysis showed that the interactive effect of the combined application of N and K fertilizers on the SPAD value of sweet potato leaves was not significant.

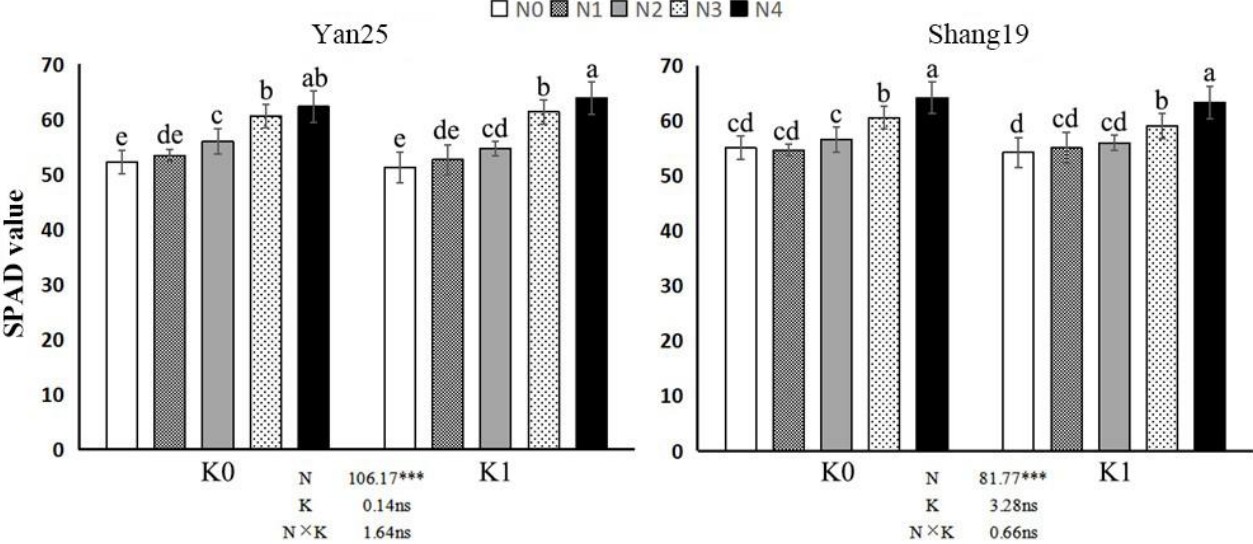

**Figure 3.** Changes in SPAD value in sweet potato leaves under different N and K levels. The letters on the column represent the significant difference at the 5% level. Columns with the same letter have no significant difference at $p < 0.05$. *** $p < 0.001$.

### 3.2.2. Leaf Photosynthetic Characteristics

Tables 2 and 3 showed the changes in Pn, Gs, Ci, and Tr in leaves of Yan25 and Shang19 at different N and K ratios. With the increase in N application, the Pn, Gs, and Ci showed a tendency of increasing and then decreasing, and reached the highest in N3 treatment, and the Tr increased continuously with the increase in N application, and reached the maximum in N4 treatment. The application of N fertilizer significantly increased the Pn, Ci and Tr of sweet potato leaves ($p < 0.05$).

**Table 2.** Effects of combined application of N and K on photosynthetic characteristics of Yan25.

| K Treatments | N Treatments | Pn/(μmol/m²·s) | Gs/(mol/m²·s) | Ci/(μmol/mol) | Tr/(mmol/m²·s) |
|---|---|---|---|---|---|
| K0 | N0 | 20.01 ± 0.99 c | 0.46 ± 0.02 bc | 256.8 ± 4.48 ef | 3.47 ± 0.42 d |
| | N1 | 20.82 ± 0.21 bc | 0.47 ± 0.04 bc | 280.1 ± 5.58 bc | 3.82 ± 0.40 cd |
| | N2 | 21.40 ± 0.72 bc | 0.48 ± 0.03 abc | 289.5 ± 3.30 ab | 4.43 ± 0.31 bc |
| | N3 | 22.68 ± 0.44 ab | 0.48 ± 0.02 abc | 298.3 ± 3.08 a | 4.92 ± 0.07 ab |
| | N4 | 21.66 ± 0.41 bc | 0.43 ± 0.03 c | 289.8 ± 2.23 ab | 5.14 ± 0.13 ab |
| K1 | N0 | 20.56 ± 0.76 bc | 0.49 ± 0.03 abc | 245.6 ± 3.35 f | 5.02 ± 0.19 ab |
| | N1 | 21.22 ± 0.65 bc | 0.53 ± 0.01 ab | 260.2 ± 2.1 de | 5.38 ± 0.17 a |
| | N2 | 22.49 ± 0.82 ab | 0.50 ± 0.01 abc | 280.7 ± 8.00 bc | 5.34 ± 0.13 a |
| | N3 | 24.76 ± 0.64 a | 0.56 ± 0.03 a | 289.6 ± 5.50 ab | 5.53 ± 0.20 a |
| | N4 | 22.37 ± 0.84 ab | 0.45 ± 0.01 bc | 273.4 ± 2.30 cd | 5.62 ± 0.22 a |
| ANOVA | | F value | | | |
| N | | 7.21 * | 2.90 ns | 27.79 *** | 41.85 *** |
| K | | 5.74 * | 6.47 * | 21.40 ** | 6.43 * |
| N × K | | 0.463 ns | 0.575 ns | 0.61 ns | 2.46 ns |

The lowercase letters in the same column indicate significant differences at the 5% level. Means followed by the same letter within a column are not significantly different at $p < 0.05$. * $p < 0.05$, ** $p < 0.01$, *** $p < 0.001$.

**Table 3.** Effects of combined application of N and K on photosynthetic characteristics of Shang19.

| K Treatments | N Treatments | Pn/(μmol/m²·s) | Gs/(mol/m²·s) | Ci/(μmol/mol) | Tr/(mmol/m²·s) |
|---|---|---|---|---|---|
| K0 | N0 | 15.51 ± 0.65 c | 0.39 ± 0.04 b | 269.67 ± 6.49 cd | 4.86 ± 0.19 e |
| | N1 | 15.88 ± 0.65 c | 0.40 ± 0.03 b | 280.4 ± 3.20 bc | 5.04 ± 0.12 de |
| | N2 | 17.30 ± 0.95 abc | 0.45 ± 0.02 ab | 288.5 ± 6.18 ab | 5.11 ± 0.12 de |
| | N3 | 19.24 ± 0.63 ab | 0.47 ± 0.04 ab | 297.6 ± 2.50 a | 5.24 ± 0.21 cde |
| | N4 | 16.77 ± 0.90 bc | 0.46 ± 0.02 ab | 274.9 ± 4.30 bcd | 5.81 ± 0.27 abc |
| K1 | N0 | 17.17 ± 0.65 abc | 0.44 ± 0.03 ab | 262.2 ± 2.40 d | 5.60 ± 0.28 bcd |
| | N1 | 17.88 ± 0.67 abc | 0.45 ± 0.02 ab | 272.3 ± 4.10 cd | 5.98 ± 0.09 ab |
| | N2 | 18.63 ± 1.11 ab | 0.48 ± 0.02 ab | 277.6 ± 3.30 bc | 6.08 ± 0.19 ab |
| | N3 | 19.59 ± 0.81 a | 0.49 ± 0.01 a | 283.2 ± 3.93 bc | 6.14 ± 0.13 ab |
| | N4 | 17.21 ± 0.40 abc | 0.47 ± 0.03 a | 271.5 ± 4.20 cd | 6.41 ± 0.19 a |
| ANOVA | | F value | | | |
| N | | 4.98 * | 3.194 ns | 9.62 ** | 5.02 * |
| K | | 5.66 * | 4.418 ns | 10.79 * | 48.18 *** |
| N × K | | 0.46 ns | 0.382 ns | 0.461 ns | 1.05 ns |

The lowercase letters in the same column indicate significant differences at the 5% level. Means followed by the same letter within a column are not significantly different at $p < 0.05$. * $p < 0.05$, ** $p < 0.01$, *** $p < 0.001$.

At the K0 level, with the increase in N application, the Pn, Gs, Ci and Tr of Yan25 leaves increased by 4.0–13.3%, 1.4–4.7%, 9.1–16.2% and 10.1–48.1%, respectively; at the K1 level, with the increase in N application, the Pn, Gs, Ci and Tr of Yan25 leaves increased by 3.2–20.4%, 2.9–14.7%, 5.9–17.9%, and 6.4–12%, respectively.

At the K0 level, with the increase in N application, the Pn, Gs, Ci, and Tr of Shang19 leaves increased by 2.4–24.1%, 2.3–22.0%, 1.9–10.3%, and 3.7–19.5%, respectively; at the K1 level, with the increase in N application, the Pn, Gs, Ci, and Tr of Shang19 leaves increased by 0.2–14.1%, 1.8–11.1%, 3.5–8.0%, and 6.8–14.5%, respectively.

K fertilization increased leaf Pn, Gs, and Tr, and reduced Ci. Compared with the K0 treatment, the Pn, Gs, and Tr of Yan25 increased by 2.0–9.2%, 3.6–15.7%, and 9.3–44.7%, respectively, and the Pn, Gs, and Tr of Shang19 after K application, increased by 1.8–12.6%, 1.2–13.2%, and 10.3–19.0%, respectively. Compared with the K0 treatment, the Ci of Yan25 and Shang19 decreased by 2.9–7.1% and 1.2–4.8%, respectively, after K application.

The two-factor analysis showed that the K fertilizer had a significant or extremely significant effect on Pn, Ci, and Tr. The combined application of the N and K fertilizer had significant effects on leaf Pn, Gs, Ci, and Tr. The interaction effect is not significant.

The Pn and Gs of the leaves of Yan25 and Shang19 reached the maximum value when the fertilizer ratio was K1N3, and the Tr reached the maximum value when K1N4, both of which were consistent with the Pn and Gs under K1N2. There was no significant difference in temperature and Tr.

### 3.3. Yield and Quality Analysis

3.3.1. Quality Analysis

The effects of N and K fertilizers on the starch, soluble sugar, and protein contents of sweet potato roots reached significant or extremely significant levels. The interaction between N and K rationing on the starch, soluble sugar, and protein contents within the roots was not significant. With the increase in N application, the starch and protein content in the roots of Yan25 and Shang19 continued to increase, while the soluble sugar content continued to decrease (Table 4).

**Table 4.** Effect of N and K combination on the nutritional ingredient of tuber.

| K Treatments | N Treatments | Yan25 | | | N Treatments | Shang19 | | |
|---|---|---|---|---|---|---|---|---|
| | | Starch/% | Soluble Sugar/% | Protein/% | | Starch/% | Soluble Sugar/% | Protein/% |
| K0 | N0 | 43.2 ± 1.06 f | 20.7 ± 0.82 ab | 3.4 ± 0.34 d | N0 | 55.4 ± 1.2 d | 12.2 ± 0.51 ab | 3.2 ± 0.07 d |
| | N1 | 45.7 ± 0.85 e | 19.1 ± 0.23 cd | 3.5 ± 0.16 cd | N1 | 59.7 ± 1.02 c | 11.5 ± 0.47 bc | 3.5 ± 0.13 d |
| | N2 | 48.2 ± 0.99 d | 17.3 ± 0.33 e | 3.9 ± 0.11 c | N2 | 62.1 ± 0.52 c | 8.9 ± 0.48 ef | 3.9 ± 0.14 cd |
| | N3 | 51.3 ± 0.57 c | 14.8 ± 0.13 g | 4.5 ± 0.10 b | N3 | 63.2 ± 0.45 c | 8.2 ± 0.82 fg | 4.7 ± 0.58 ab |
| | N4 | 52.2 ± 0.99 c | 14.3 ± 0.23 g | 4.8 ± 0.14 ab | N4 | 64.1 ± 1.37 c | 7.2 ± 0.12 g | 5.1 ± 0.06 a |
| K1 | N0 | 48.7 ± 0.53 d | 21.6 ± 0.58 a | 3.4 ± 0.12 d | N0 | 62.8 ± 1.78 c | 13.0 ± 0.25 a | 3.8 ± 0.12 cd |
| | N1 | 51.3 ± 0.61 c | 20.0 ± 0.43 bc | 3.6 ± 0.18 cd | N1 | 63.6 ± 0.33 c | 11.6 ± 0.33 bc | 4.3 ± 0.46 bc |
| | N2 | 54.6 ± 0.60 b | 18.7 ± 0.18 d | 4.0 ± 0.12 c | N2 | 67.5 ± 0.65 b | 10.4 ± 0.42 cd | 4.6 ± 0.03 abc |
| | N3 | 59.7 ± 0.46 a | 17.6 ± 0.35 e | 4.9 ± 0.07 a | N3 | 70.3 ± 0.26 a | 10.3 ± 0.42 d | 4.9 ± 0.31 ab |
| | N4 | 60.1 ± 0.73 a | 15.8 ± 0.38 f | 5.1 ± 0.12 a | N4 | 71.2 ± 0.65 a | 9.5 ± 0.33 de | 5.3 ± 0.13 a |
| ANOVA | | | | | F value | | | |
| N | | 118.1 *** | 132.2 *** | 125.9 *** | N | 143.0 *** | 44.8 *** | 17.16 *** |
| K | | 358.4 *** | 60.7 *** | 7.7 * | K | 383.8 *** | 35.6 *** | 12.17 ** |
| N × K | | 3.2 ns | 3.1 ns | 3.2 ns | N × K | 3.2 ns | 3.2 ns | 2.2 ns |

The lowercase letters in the same column indicate significant differences at the 5% level. Means followed by the same letter within a column are not significantly different at $p < 0.05$. * $p < 0.05$, ** $p < 0.01$, *** $p < 0.001$.

At the K0 level, with the increase in N application, the starch content and protein content of Yan25 increased by 5.8–20.8% and 4.0–42.0%, respectively, and the soluble sugar content decreased by 7.5–31.0%; at the K1 level, with the increase in N application, the starch content and protein content of Yan25 increased by 5.4–24.5% and 2.8–46.5%, respectively, and the soluble sugar content decreased by 7.4–27.2%.

At the K0 level, with the increase in N application, the starch content and protein content of Shang19 increased by 7.8–15.8% and 8.5–59.3%, respectively, and the soluble sugar content decreased by 5.3–40.5%; at the K1 level, with the increase in N application, the starch content and protein content of Shang19 increased by 4.6–17.2% and 12.5–38.5%, respectively, and the soluble sugar content decreased by 10.3–26.4%.

Compared with the K0 treatment, after K application, the starch, soluble sugar, and protein contents of Yan25 increased by 12.2–16.4%, 4.6–18.9%, and 1.1–10.3%, respectively, and the starch, soluble sugar, and protein contents of Shang19 increased by 6.5–11.2%, 0.9–31.9%, and 19.3–65.2%, respectively. The materials of this study, Yan25 and Shang19, had the highest starch and protein contents at fertilizer ratios of K1N4, and the highest soluble sugar content at K1N0.

### 3.3.2. Yield Analysis

As shown in Tables 5 and 6, the effects of the N fertilizer on the number of tubers per plant, fresh weight of the roots, dry weight of the roots, and vine length of the sweet potato reached highly significant levels. The effects of the K fertilizer on the number of potatoes per plant, fresh weight of the roots, dry weight of the roots, and stem diameter reached significant or highly significant levels, and the effects of the N and K interaction on the root fresh weight, dry rate, and root dry weight reached significant or extremely significant

levels. As the amount of N application continued to increase, the number of tubers per plant, fresh root weight, and dry root weight of Yan25 and Shang19 first increased and then decreased, and the vine length continued to increase.

**Table 5.** Effects of combined application of N and K on yield and yield components of Yan25.

| Varieties | K Treatments | N Treatments | The Number of Tubers Per Plant | Root Fresh Weight/(t/ha$^{-1}$) | Dry Rate/% | Root Dry Weight/(t/ha$^{-1}$) | Vine Length/mm | Stem Diameter/mm |
|---|---|---|---|---|---|---|---|---|
| Yan25 | K0 | N0 | 3.11 ± 0.42 c | 31.08 ± 1.03 d | 28.12 ± 0.40 bc | 8.73 ± 0.16 fg | 272.83 ± 3.93 e | 11.55 ± 0.11 c |
| | | N1 | 3.89 ± 0.42 bc | 34.20 ± 0.95 c | 29.79 ± 0.23 abc | 10.19 ± 0.21 de | 276.67 ± 5.33 e | 11.63 ± 0.21 bc |
| | | N2 | 4.67 ± 0.27 ab | 37.80 ± 1.02 b | 27.65 ± 1.50 c | 10.50 ± 0.29 cd | 296.50 ± 5.20 d | 12.06 ± 0.29 abc |
| | | N3 | 4.33 ± 0.47 ab | 37.20 ± 0.37 b | 30.63 ± 1.07 ab | 11.40 ± 0.51 bc | 334.33 ± 9.03 bc | 12.45 ± 0.47 abc |
| | | N4 | 3.67 ± 0.27 bc | 33.53 ± 0.68 c | 28.77 ± 0.91 abc | 9.64 ± 0.11 def | 363.67 ± 9.67 a | 12.07 ± 0.49 abc |
| | K1 | N0 | 4.00 ± 0.54 bc | 28.80 ± 0.55 d | 28.62 ± 0.23 abc | 8.24 ± 0.22 g | 252.50 ± 6.91 f | 11.96 ± 0.10 bc |
| | | N1 | 4.44 ± 0.45 bc | 33.83 ± 0.45 c | 27.34 ± 0.74 c | 9.25 ± 0.13 ef | 268.50 ± 5.33 ef | 12.51 ± 0.28 abc |
| | | N2 | 5.22 ± 0.31 a | 39.13 ± 0.92 ab | 30.41 ± 0.78 ab | 11.9 ± 0.59 ab | 297.00 ± 1.73 d | 12.55 ± 0.33 abc |
| | | N3 | 5.11 ± 0.57 a | 40.73 ± 0.42 a | 30.84 ± 0.21 a | 12.56 ± 0.21 a | 316.67 ± 3.17 c | 12.75 ± 0.14 ab |
| | | N4 | 4.44 ± 0.42 ab | 39.37 ± 0.66 ab | 31.16 ± 0.12 a | 12.27 ± 0.16 ab | 336.67 ± 3.53 b | 12.92 ± 0.50 a |
| | ANOVA | | | | F value | | | |
| | | N | 6.26 ** | 49.89 *** | 3.50 * | 41.58 *** | 76.51 *** | 2.21 ns |
| | | K | 13.13 ** | 10.98 * | 2.04 ns | 15.63 ** | 15.21 ** | 8.22 * |
| | | N × K | 0.212 ns | 9.18 ** | 3.80 * | 11.75 ** | 1.68 ns | 0.33 ns |

The lowercase letters in the same column indicate significant differences at the 5% level. Means followed by the same letter within a column are not significantly different at $p < 0.05$. * $p < 0.05$, ** $p < 0.01$, *** $p < 0.001$.

**Table 6.** Effects of combined application of N and K on yield and yield components of Shang19.

| Varieties | K Treatments | N Treatments | The Number of Tubers Per Plant | Root Fresh Weight/(t/ha$^{-1}$) | Dry Rate/% | Root Dry Weight/(t/ha$^{-1}$) | Vine Length/mm | Stem Diameter/mm |
|---|---|---|---|---|---|---|---|---|
| Shang19 | K0 | N0 | 3.33 ± 0.27 e | 26.10 ± 0.79 de | 35.42 ± 0.60 cd | 9.24 ± 0.12 e | 236.33 ± 3.87 d | 12.17 ± 0.61 c |
| | | N1 | 4.00 ± 0.27 d | 30.54 ± 0.48 bc | 35.73 ± 0.21 c | 10.91 ± 0.11 c | 291.33 ± 2.33 c | 12.48 ± 0.20 c |
| | | N2 | 4.67 ± 0.27 bc | 30.53 ± 0.66 bc | 39.15 ± 0.65 a | 12.05 ± 0.16 b | 293.00 ± 4.20 c | 13.36 ± 0.27 c |
| | | N3 | 4.11 ± 0.16 cd | 30.84 ± 0.63 b | 38.11 ± 0.22 ab | 11.75 ± 0.17 b | 402.00 ± 7.00 a | 13.63 ± 0.40 bc |
| | | N4 | 3.78 ± 0.16 de | 27.54 ± 0.23 de | 36.79 ± 0.10 bc | 10.13 ± 0.11 cd | 405.00 ± 6.78 a | 15.10 ± 0.97 b |
| | K1 | N0 | 4.11 ± 0.16 cd | 25.33 ± 0.99 e | 37.54 ± 0.79 b | 9.36 ± 0.02 de | 254.33 ± 2.33 d | 12.38 ± 0.30 c |
| | | N1 | 4.89 ± 0.16 ab | 28.18 ± 0.95 cd | 37.55 ± 0.48 b | 10.58 ± 0.33 c | 297.33 ± 7.87 c | 12.69 ± 0.26 c |
| | | N2 | 5.33 ± 0.54 a | 36.25 ± 1.01 a | 34.12 ± 0.66 de | 12.37 ± 0.39 ab | 306.00 ± 7.20 c | 17.40 ± 0.49 a |
| | | N3 | 4.67 ± 0.27 bc | 37.59 ± 1.05 a | 36.63 ± 0.63 bc | 13.12 ± 0.54 a | 348.33 ± 11.13 b | 13.74 ± 0.60 bc |
| | | N4 | 4.11 ± 0.16 cd | 37.21 ± 0.38 a | 33.78 ± 0.23 e | 12.57 ± 0.19 ab | 348.75 ± 6.55 b | 15.11 ± 0.13 b |
| | ANOVA | | | | F value | | | |
| | | N | 13.67 *** | 41.34 *** | 5.20 * | 47.43 *** | 125.87 *** | 12.31 ** |
| | | K | 29.00 *** | 61.34 *** | 14.35 ** | 24.18 ** | 12.69 * | 5.92 * |
| | | N × K | 0.64 ns | 22.50 *** | 21.70 *** | 9.25 ** | 44.75 *** | 7.20 * |

The lowercase letters in the same column indicate significant differences at the 5% level. Means followed by the same letter within a column are not significantly different at $p < 0.05$. * $p < 0.05$, ** $p < 0.01$, *** $p < 0.001$.

At the K0 level, with the increase in N application, the number of tubers per plant, root fresh weight, root dry weight, and vine length of Yan25 increased by 17.9–50.0%, 7.9–21.6%, 10.4–36.3%, and 1.4–33.3%, respectively; at the K1 level, with the increase in N application, the number of tubers per plant, root fresh weight, root dry weight, and vine length of Yan25 increased by 11.1–30.6%, 17.5–41.4%, 12.2–52.5%, and 6.3–33.3%, respectively.

At the K0 level, with the increase in N application, the number of tubers per plant, root fresh weight, root dry weight, and vine length of Shang19 increased by 13.3–40.0%, 5.5–18.1%, 9.6–30.4%, and 23.3–71.4%, respectively; at the K1 level, with the increase in N application, the number of tubers per plant, root fresh weight, root dry weight, and vine length of Shang19 increased by 0–29.7%, 11.3–48.4%, 13.1–35.8%, and 16.9–37.1%, respectively.

K fertilizer application significantly increased the number of tubers per plant and stem diameter of sweet potato. Compared with the K0 treatment, the number of tubers per plant and stem diameter of Yan25 increased by 9.52–28.57% and 2.48–7.60% after K application. Compared with the K0 treatment, after K application, the number of tubers per plant and stem diameter of Shang19 increased by 8.82–23.33% and 0.07–30.27%, respectively. K application at low N (N0 and N1) reduced the fresh weight and dry weight of the roots, and K application at N2–N4 significantly increased the fresh weight and dry weight of the roots.

The application of K at N3 and N4 reduced the length of the sweet potato vines, inhibited the excessive growth of the aboveground parts, and increased the root tuber yield.

## 4. Discussion

### 4.1. Effects of Combined Application of N and K on Metabolic Enzymes of Sweet Potato Leaves

A large number of studies have shown that GS is the key enzyme for N assimilation and has an important impact on protein synthesis [29,30]. The increase in N fertilizer significantly increases the activity of GS, and applying a certain amount of N fertilizer is very important for N metabolism [31]. Jiao et al. [32] found that the application of a certain amount of K fertilizer would increase GS activity and have the effect of promoting N metabolism. This experiment was consistent with the results of previous studies. The K1N2 treatment effectively increased the GS activity of sweet potato leaves, and excessive N fertilizer reduced the GS activity. Soluble protein plays an important role in the process of assimilation metabolism. The soluble protein content can reflect the N metabolism level of the plant and the viability of the leaves [30]. Geng et al. [33] found that, with the increase in N application, the soluble protein content in plant leaves first increased and then decreased. This was consistent with the results of this study. When the N application rate was $120 \, \text{kg} \cdot \text{ha}^{-1}$, the soluble protein reached the maximum. As the N application rate further increased, the soluble protein decreased. This was attributed to the inhibition of metabolic activities due to premature cell senescence in the leaves caused by high N.

This study showed that the application of K fertilizer would significantly increase the soluble protein content in the leaves, due to the ability of K to promote the absorption and utilization of N in sweet potato, and the increase in N content in the leaves, which is conducive to the formation of soluble proteins.

### 4.2. Effects of Combined Application of N and K on Photosynthetic Characteristics of Sweet Potato

The N fertilizer affects the production and accumulation of dry matter by promoting leaf growth and improving photosynthetic efficiency [34]. Photosynthesis is the main pathway for plant material metabolism and energy conversion, which directly affects crop growth conditions and is the basis for crop yield formation [35]. Wang et al. [36] showed that the SPAD value indirectly reflect the chlorophyll content of plant leaves. N is a key substrate for chlorophyll and protein synthesis, directly affecting the formation of chlorophyll and the absorption and transport of carbon dioxide. A high chlorophyll content indicates more efficient photosynthesis, and plants can synthesize organic matter more efficiently, thereby maintaining healthy growth [37,38]. The increased application of the N fertilizer can improve the photosynthetic performance of potato leaves [39,40]. Wei et al. [41] showed that the effect of the N fertilizer on the photosynthesis of leafy sweet potato. Applying N fertilizer can increase the leaf chlorophyll content, and plant Pn, Gs, and Tr. Studies in crops such as potatoes [42] and corn [43] have found that applying K fertilizer can increase the Pn, Gs, and Tr in leaves, and reduce the Ci.

This study is consistent with the results of previous studies. N and K fertilizers can increase the SPAD value, net photosynthetic rate, stomatal conductance, and transpiration rate of leaves. K application reduces the intercellular $CO_2$ concentration in leaves. This shows that combined N and K fertilization can significantly increase the photosynthetic rate, while intercellular $CO_2$ is rapidly utilized, resulting in a decrease in intercellular $CO_2$ concentration in the leaves. The experimental results show that, when the appropriate amount of N and K is applied (K1N2), the photosynthetic performance of sweet potato leaves is the best, but excessive N fertilizer (K1N3 and K1N4) will lead to an excessively large leaf area index and canopy shading, thereby reducing the efficiency of photosynthesis.

### 4.3. Effects of Combined Application of N and K on Nutrient Composition and Yield of Sweet Potato Roots

N fertilizer has an important impact on the yield and quality of sweet potato. The effect of N on quality mainly depends on root carbon and N metabolism and related enzyme

activities, which can improve the sweet potato root quality to a certain extent [43]. N can increase the starch rate of sweet potato roots and is beneficial to starch accumulation [44]. The stem is the main organ transporting photosynthetic products from the source to the sink [45]. Increasing K fertilizer can increase the stem thickness of sweet potato, improve stem transportation efficiency, facilitate the transport of photosynthetic products to the tubers, and promote the synthesis and accumulation of soluble sugar, protein, and starch in the tubers, thereby increasing the sweet potato yield [46,47]. N fertilization significantly increased the protein content in sweet potato roots [48]. Yao et al. [49] showed that the soluble sugar content under no N application treatment was higher than that under N application treatment. Applying K fertilizer can increase the soluble sugar content of plants [50,51], which is consistent with the results of this study. Hou et al. [52] and others found, in their research on edible sweet potato, that the application of K fertilizer could increase the protein content in the roots. Research results on the effect of K fertilizer on protein in plants varied. In this study, under the same N conditions, the application of K fertilizer increased the protein content in sweet potato roots, but there was no significant difference in the protein content of K0 and K1. This study showed that the starch and protein contents in sweet potato roots were the highest under the K1N4 treatment, and the soluble sugar content in the roots was the highest under the K1N0 treatment. The appropriate application of N will increase the number of tubers per sweet potato plant and increase the yield [53]. Wu et al. [54] showed that the number of tubers increased first and then decreased after applying the N fertilizer. Increasing the application of K fertilizer is one of the effective means to increase the sweet potato yield [55]. K fertilization benefits the movement of photosynthetic products from aboveground to underground, provides sufficient material basis for root expansion, and promotes the growth of sweet potato dry matter above and below ground [56]. Wang et al. [57] further found that K application can increase the weight of sweet potato tubers and increase the number of tubers, thereby significantly increasing the yield of sweet potato.

This study was basically consistent with previous studies. Applying a certain amount of N fertilizer increased the number of tubers per plant, fresh weight, and dry weight. When no K fertilizer is applied, the N application rate is 120 kg·ha$^{-1}$ (N2). The fresh weight and dry root weight reach the maximum value. When the K fertilizer is applied at 240 kg·ha$^{-1}$, the N application amount is 180 kg·ha$^{-1}$ (N3), and the fresh weight and dry weight of roots reached the maximum; the application of K fertilizer significantly increased the number of tubers per plant, and the fresh weight and dry weight of roots, which may be because K fertilizer promotes the absorption of N fertilizer by sweet potato. Wang et al. [58] showed that, as the amount of N application increases, vine growth continues to increase, and increasing K fertilizer can inhibit the aboveground growth caused by high N to a certain extent. This study shows that, under the condition of appropriate amounts of N fertilizer and K fertilizer (K1N3), the fresh weight and dry weight of sweet potato roots reach the maximum value, and there is no significant difference from K1N2. However, excessive N fertilizer will lead to the excessive growth of the aboveground part, thus reducing the yield of tubers, while the appropriate application of K fertilizer can inhibit this negative effect and increase the overall yield. This study was conducted under specific soil and climate conditions, while the key factors affecting the sweet potato yield (such as nutrient requirements, water management, root development, etc.) have similar physiological mechanisms under different environments, so the research results are still scalable. In summary, optimizing the yield by adjusting water and fertilizer strategies still applies to different soil fertility and climate conditions.

*4.4. Effects of Photosynthetic Characteristics of Sweet Potato on Root Tuber Yield and Quality under N and K Combination Conditions*

Photosynthesis is the process by which plants absorb light energy through their leaves and convert carbon dioxide and water into organic matter. These organic compounds are not only used for the growth and development of the plant but also transferred to

the underground tuberous roots, affecting their yield and quality [59]. Sweet potato has a typical sink–source relationship. Photosynthesis in the aboveground part produces a large number of photosynthetic products, which are transported to the underground tubers through the phloem for their growth and expansion [16].

Adequate N contributes to the synthesis of chlorophyll, provides energy and reducing power for carbon assimilation, and increases stomatal conductance and mesophyll thickness, improving the conduction of $CO_2$ in leaves [60,61]. K in plants mainly improves the photosynthetic capacity of plants by maintaining the balance of cell osmotic pressure, increasing the activity of key photosynthetic enzymes, promoting stomatal movement and the transport of photosynthetic products [13]. Wang et al. [24] found that N increases the accumulation of photosynthetic products mainly by increasing the photosynthetic activity, and K fertilizer can increase the activity of carbon metabolism enzymes and accelerate the transport of photosynthetic products. This study is basically consistent with previous studies. With the increase in N application, the vine length and stem diameter continue to increase. K application is beneficial to increasing the stem diameter and reducing the vine length under high N conditions, shortening the source–sink distance, increasing the transport rate of photosynthetic products, and promoting the accumulation of the starch, protein, and soluble sugar content in tubers. This study shows that the photosynthetic performance of sweet potato leaves and tuber yield reach the optimal level at K1N2.

This study selected two different types of sweet potato varieties for the experiment. Although the varieties have some limitations, the effects of N and K fertilizers on sweet potato growth, development, and physiological effects were the same. Therefore, these results may also show similar trends for other varieties. However, the degree of response may vary depending on the growth habits and specific nutrient requirements of different varieties.

## 5. Conclusions

This study found that a certain amount of K fertilizer could effectively increase the activity of GS, promote N metabolism, and then improve the utilization rate of the N fertilizer when the N application rate was 120 kg·ha$^{-1}$. The application of appropriate amounts of N and K fertilizers can improve the photosynthetic characteristics of sweet potato leaves and increase the accumulation of photosynthetic products. These photosynthetic products are transported to the root tubers through the stems, promoting the expansion of the root tubers and the accumulation of nutrients, improving the yield and quality of the root tubers. For farmers, the appropriate combined application of N and K is an effective strategy to optimize sweet potato growth and maximize the yield. This fertilization method not only improves the utilization efficiency of the N fertilizer but also improves the photosynthetic capacity of the plant, thereby increasing the yield and quality of the tubers. However, excessive N application can lead to an excessive leaf area index, reduced photosynthetic efficiency, and excessive vegetative growth, which has a negative impact on tuber development. K fertilizer is key in balancing vegetative growth and ensuring optimal yield and quality. Therefore, in actual implementation, farmers should focus on the balance and moderate application of N and K fertilizers to achieve the best results in sweet potato cultivation.

**Author Contributions:** Writing—original draft, X.S.; writing—review and editing, X.S., M.J., J.Y. and Z.L.; investigation, X.S., M.J., S.W., X.X., L.D., Z.Z., X.Z., Y.Z. and G.L.; validation, S.W. and X.X.; methodology, L.D. and Z.Z.; data curation, X.Z. and Z.L.; formal analysis, J.Y.; project administration, Y.Z. and G.L.; conceptualization, Z.L.; funding acquisition, Z.L. All authors have read and agreed to the published version of the manuscript.

**Funding:** This work was supported by the Natural Science Foundation of China (32071897, 32272222), the Three Rural Areas and Nine Rural Areas of Zhejiang Province (2022SNJF007), and the Ningbo Key Projects (2022S092).

**Data Availability Statement:** All data generated or analyzed during this study are available within the article or upon request from the corresponding author.

**Conflicts of Interest:** The authors declare that they have no known competing financial interests or personal relationships that could have appeared to influence the work reported in this paper.

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
