# Peer review of "The Effect of Nitrogen and Potassium Interaction on the Leaf Physiological Characteristics, Yield, and Quality of Sweet Potato"

_agronomy, doi:10.3390/agronomy14102319_

Round 1

Reviewer 1 Report

Comments and Suggestions for Authors

The manuscript “The effect of nitrogen and potassium interaction on the leaf physiological characteristics and yield and quality of sweet potato during the root expansion period” deals with the critical issue in sweet potato cultivation by exploring the effects of nitrogen (N) and potassium (K) fertilization on the physiological characteristics, yield, and quality of sweet potatoes.

Comments

·       It would be beneficial to explain why these varieties were chosen and how representative they are of sweet potato cultivation more broadly in the material and method section.

·       The study uses only two potassium levels (K0 and K1), which limits the exploration of K’s effects. Including intermediate K levels could provide a more comprehensive understanding.

·       While SPAD value increases with N application, the study should discuss how this correlates with chlorophyll content and overall plant health.

·       How did the application of K fertilizer influence the partitioning of nutrients (e.g., N, C) between the leaves and tubers?

·       LN 102-108: Please rewrite with better clarity.

·       The study mentions K significantly affects photosynthesis parameters, but does not delve into the specific physiological roles of K in enhancing photosynthesis.

·       How do the researchers plan to translate these findings into practical recommendations for farmers, especially in terms of optimizing N and K application rates?

·       Section 2.4. Statistical Analysis section should be written in detail. It seems that authors have used Graph Pad for making figures. This should be written in Statistical Analysis section.

·       The observed trends in yield components (e.g., tuber number, fresh weight) should be related to potential agronomic practices for optimizing sweet potato production.

·       The study should discuss the scalability of its findings to different agricultural contexts, particularly in regions with varying soil fertility and climatic conditions.

·       How reproducible are the results of this study in different geographical regions or under varying climatic conditions?

·       The findings are based on two varieties; the study should discuss whether the results are likely to be applicable to other sweet potato varieties with different growth habits or nutrient requirements.

·       What future research directions are suggested by this study, particularly in terms of optimizing fertilizer application for different sweet potato varieties or under different environmental conditions?

Comments on the Quality of English Language

 Minor editing of English language required.

Author Response

Comments 1: It would be beneficial to explain why these varieties were chosen and how representative they are of sweet potato cultivation more broadly in the material and method section.

Response 1: Thank you for your valuable suggestions. Based on your suggestions, we have made modifications and additions in the discussion section, as follows:

Line 105-107: In this study, starchy sweet potato ‘Shang19’ and fresh sweet potato ‘Yan25’ were selected as experimental materials. Both varieties have been widely promoted, which have earned high market recognition and high economic value.

Comments 2: The study uses only two K levels (K0 and K1), which limits the exploration of K’s effects. Including intermediate K levels could provide a more comprehensive understanding.

Response 2: Thank you for your valuable suggestions. Including an intermediate K level could indeed provide a more comprehensive understanding of the effect of K on sweet potato growth and N absorption. However, in this study, we selected two K levels, K0 and K1, with the aim of highlighting the contrast between K supply and deficiency under extreme conditions. This design is able to more clearly reveal the effects of K on sweet potato growth and N uptake. Nevertheless, we also recognize that the lack of intermediate K levels may indeed limit the in-depth exploration of K effects. In future studies, we will consider adding intermediate K levels to more accurately depict the range of K effects and the optimal K application rate in order to obtain more comprehensive conclusions.

Comments 3. While SPAD value increases with N application, the study should discuss how this correlates with chlorophyll content and overall plant health.

Response 3: We would like to thank you for pointing this out. According to your suggestion, we have modified and supplemented page 12 of the Discussion section as follows:

Line 348-353: Wang et al [36] showed that the SPAD value indirectly reflect the chlorophyll content of plant leaves. N is a key substrate for chlorophyll and protein synthesis, directly affecting the formation of chlorophyll and the absorption and transport of carbon dioxide. High chlorophyll content indicates more efficient photosynthesis, and plants can synthesize organic matter more efficiently, thereby maintaining healthy growth [37,38].

Comments 4. How did the application of K fertilizer influence the partitioning of nutrients (e.g., N, C) between the leaves and tubers?

Response 4: Thank you for your insightful question. In the revised manuscript, the corresponding explanation and conclusion have been added which can be found in page 12 and given below:

Line 375-380: The stem is the main organ transporting photosynthetic products from the source to the sink [45]. Increasing K fertilizer can increase the stem thickness of sweet potato, im-prove stem transportation efficiency, facilitate the transport of photosynthetic products to the tubers, and promote the synthesis and accumulation of soluble sugar, protein, and starch in the tubers, thereby increasing sweet potato yield [46,47].

Comments 5. LN 102-108: Please rewrite with better clarity.

Response 5: Thank you for your valuable suggestions for the manuscript. We have made the following revisions to your questions:

Line 96-100: This experiment was conducted in Guantang, Zhejiang A&F University (119°73′E, 30°23′N) during the 2021. The area comes under subtropical monsoon climate, warm and humid environment with sufficient light, abundant rainfall during rainy season. The average annual precipitation is 1613.9 mm, with 158 rainy days and an average annual frost-free period of 237 days.

Comments 6. The study mentions K significantly affects photosynthesis parameters, but does not delve into the specific physiological roles of K in enhancing photosynthesis.

Response 6: Thank you for your professional suggestion. According to your suggestion, we have added the discussion in Section 4.4 on page 13.

Line 432-435: K in plants mainly improves the photosynthetic capacity of plants by maintaining the balance of cell osmotic pressure, increasing the activity of key photosynthetic enzymes, promoting stomatal movement and the transport of photosynthetic products [13].

Comments 7. How do the researchers plan to translate these findings into practical recommendations for farmers, especially in terms of optimizing N and K application rates?

Response 7: Thank you for your insightful questions about the research content. We have responded to your questions as follows:

Based on the study's results, it is evident that a balanced combination of N and K fertilizers is key to optimizing sweet potato growth, yield, and quality. Therefore, we recommend that farmers avoid excessive N fertilizer application and apply an appropriate amount of K fertilizer at the same time to promote the N absorption capacity of sweet potatoes, increase yield and obtain high returns.

 Comments 8. Section 2.4. Statistical Analysis section should be written in detail. It seems that authors have used Graph Pad for making figures. This should be written in Statistical Analysis section.

Response 8: We greatly appreciate your valuable suggestions. According to your suggestions, we have made the following supplements and explanations in the statistical analysis section.

Line 164-168: Microsoft office excel 2016 is used for data entry, organization and preliminary calculation. The data were analyzed by two-way ANOVA and Duncan's multiple range test (P < 0.05) using SPSS 22.0 to evaluate the effects of N and K treatments and their interactions on various physiological parameters of sweet potato plants. GraphPad Prism9.5 drawing.

 Comments 9. The observed trends in yield components (e.g., tuber number, fresh weight) should be related to potential agronomic practices for optimizing sweet potato production.

Response 9: Thank you for raising important questions about the manuscript. We have made revisions to the Conclusion section to address your concerns.

Line 459-467: For farmers, the appropriate combined application of N and K is an effective strategy to optimize sweet potato growth and maximize yield. This fertilization method not only improves the utilization efficiency of N fertilizer but also improves the photosynthetic capacity of the plant, thereby increasing the yield and quality of tubers. However, excessive N application can lead to excessive leaf area index, reduced photosynthetic efficiency, and excessive vegetative growth, which has a negative impact on tuber development. K fertilizer is key in balancing vegetative growth and ensuring optimal yield and quality. Therefore, in actual implementation, farmers should focus on the balance and moderate application of N and K fertilizers to achieve the best results in sweet potato cultivation.

Comments 10. The study should discuss the scalability of its findings to different agricultural contexts, particularly in regions with varying soil fertility and climatic conditions.

Response 10: Thank you for putting forward these important views. Based on your suggestions, we have made the following additions to our discussion of the yield analysis section:

Line 415-420: This study was conducted under specific soil and climate conditions, while the key factors affecting sweet potato yield (such as nutrient requirements, water management, root development, etc.) have similar physiological mechanisms under different environments, so the research results are still scalable. In summary, optimizing yield by adjusting water and fertilizer strategies still applies to different soil fertility and climate conditions.

Comments 11. How reproducible are the results of this study in different geographical regions or under varying climatic conditions?

Response 11: Thanks for your good questions. The results of this study are based on specific environmental and experimental conditions. Although the physiological responses of sweet potato to N and K may be consistent due to the general role of N and K in plant metabolism, the extent of these effects may vary in different geographical regions or climatic conditions. Factors such as soil type, temperature, rainfall, and sunshine may affect the response of sweet potato plants to fertilization.For example, in areas with different soil nutrient status, sweet potatoes may need adjusted N or K levels for optimal growth. In areas with different climatic conditions, the balance between plant growth and tuber development may change.Therefore, although the general trends observed in the study should be reproducible, N and K application rates should be fine-tuned according to local conditions when repeating experiments in different regions.

Comments 12. The findings are based on two varieties; the study should discuss whether the results are likely to be applicable to other sweet potato varieties with different growth habits or nutrient requirements.

Response 12: Thank you for the comments and suggestions on the manuscript. Based on this suggestion, we have supplemented and explained on page 14 of the manuscript.

Line 445-450: This study selected two different types of sweet potato varieties for the experiment. Although the varieties have some limitations, the effects of N and K fertilizers on sweet potato growth, development, and physiological effects were the same. Therefore, these results may also show similar trends for other varieties. However, the degree of response may vary depending on the growth habits and specific nutrient requirements of different varieties.

Comments 13. What future research directions are suggested by this study, particularly in terms of optimizing fertilizer application for different sweet potato varieties or under different environmental conditions?

Response 13: We are very grateful for this thought-provoking suggestion, and in response we offer the following perspectives for future research.

The findings highlight several key areas for further research, particularly in optimizing fertilizer application across different sweet potato varieties and under varying environmental conditions.

  1. Varietal differences: Because this study focused on two specific sweet potato varieties, future research should examine how different varieties with varying growth habits, rates of tuber development, and nutrient requirements respond to N and K fertilizers. This will help to tailor fertilizer recommendations for a wider range of cultivars.
  2. Environmental differences: This study was conducted under specific climatic and soil conditions. Future research should explore how different environmental factors affect the sweet potato’s response to N and K fertilization. Conducting field trials in different geographical areas will help improve fertilization strategies based on local climatic and soil conditions.
  3. Interactions with other nutrients: While this study focused on N and K, future research could expand to investigate how other essential nutrients, such as phosphorus or micronutrients, interact with N and K fertilization. This would provide a more complete understanding of sweet potato's nutrient requirements and further refine fertilizer management practices.

By addressing these areas, future research could improve the precision and adaptability of fertilizer recommendations to ensure that sweet potato yields and quality are optimized across varieties and environmental conditions.

Reviewer 2 Report

Comments and Suggestions for Authors

Dear Authors,

COMMENTS- agronomy-3177058

The overuse or imbalance application of chemical fertilizers (N, P, K) to achieve higher productivity of crops may cause soil degradation, as well as increase the cost of cultivation. There is a growing need to explore the corrective dose of major nutrients for better crop productivity with super-quality produce without adverse effects on the field soil. The manuscript has the determined goal of providing an important way for sweet potato producers to maintain better productivity and tuber root quality by the application of appropriate doses of nitrogen and potassium chemical fertilizer. It's good to see that the authors are working on the required quantity of nutrients to save additional fertilizer wastage and reduce the cost of cultivation. 

1. Research TitleThe title of the manuscript is informative and appropriate, but requires minor editing.

The effect of nitrogen, potassium, and their interactions on the leaf physiological characteristics, yield, and quality of sweet potato during the root expansion period”. 

2. Abstract: The abstract part was written appropriately; this section requires many corrections and upgrades.  Rewrite this section as per the enclosed PDF.

3. IntroductionRequired minor editing as well as incorporation of research hypotheses. 

4. Materials and MethodsIt is sufficient and scientifically presented, but requires minor corrections.

5. Tables: The tables appropriately present the results.    

6. Figures: Nicely presented.

7. Statistical Analysis:  Experimental statistical design and graphical presentation is appropriate

9. Results: The results of the experiment were properly presented with the help of tables and figures.  

 10. Discussion: Experimental discussion was extremely supported by closely related studies, but summarization was also required.   

11. ConclusionIt is adequately presented.

12. Reference:  Adequately presented, required minor corrections and advice to follow the journal format for citations and references. Also, reduce the number of references up to 40 to 45.

13. English language: Required minor corrections.  

Comments on the Quality of English Language

Minor editing required 

Author Response

Thank you for your thorough review and constructive feedback on our manuscript. We have carefully considered all of your suggestions and have implemented the recommended changes, including the improvements to sentence structure and the correction of errors. Your attention to detail has greatly contributed to enhancing the clarity and overall quality of the manuscript. We sincerely appreciate your time and effort, and we believe the revisions have strengthened our work.

Reviewer 3 Report

Comments and Suggestions for Authors

Dear authors, i read with interest the manuscript "The effect of nitrogen and potassium interaction on the leaf physiological characteristics and yield and quality of sweet potato during the root expansion period". I think that the article is interesting but some major comments are reported in the attached PDF. I suggest the authors to follow the comments, specifically related to the M&M section. Results and discussion well done.

Author Response

Comments 1. please reduce the long of the title.

Response 1: Thank you for your professional suggestion. According to your suggestion, we have modified the article title:

The original title: The effect of nitrogen and potassium interaction on the leaf physiological characteristics and yield and quality of sweet potato during the root expansion period

The modified title: The effect of nitrogen and potassium interaction on the leaf physiological characteristics, yield and quality of sweet potato

Comments 2. add two references:

https://doi.org/10.3390/plants12061319

https://doi.org/10.3390/biology12020266

Response 2: Thank you for your introduction to these wonderful research work. According to your suggestion, we properly cite these articles as:

Line 519-524:

  1. Lamaro, G.P.; Tsehaye, Y.; Girma, A.; Vannini, A.; Fedeli, R.; Loppi, S. Evaluation of Yield and Nutraceutical Traits of Orange-Fleshed Sweet Potato Storage Roots in Two Agro-Climatic Zones of Northern Ethiopia. Plants 2023, 12, 13
  2. Lamaro, G.P.; Tsehaye, Y.; Girma, A.; Vannini, A.; Fedeli, R.; Loppi, S. Essential Mineral Elements and Potentially Toxic Elements in Orange-Fleshed Sweet Potato Cultivated in Northern Ethiopia. Biology 2023, 12, 266.

Comments3. please add a figure of the experimental site.

Response 3: Thank you for your suggestion to include figure of the experimental field. Unfortunately, due to equipment limitations during the data collection period, the available photographs do not meet the required resolution and clarity for publication. We sincerely apologize for this constraint and appreciate your understanding. The following are photos of the test site taken at that time:

Comments 4. “The soluble protein content was measured using the Coomassie Brilliant Blue G-250 colorimetric method; the GS activity was measured using the FeCl3 colorimetric method.” please explicit the methods.

Response 4: Thank you for your valuable suggestions. Based on your suggestions, we have added to the materials and methods section.

Line 120-127: First, the enzyme solution was extracted from 1.0 g sweet potato leaves with 3 ml ex-traction buffer. Next, 0.7 ml of the enzyme solution was combined with 1.6 ml of the reaction mixture and 0.7 ml of ATP solution, mixed thoroughly, and insulated at 37 °C for 30 minutes. Then, 1.0 ml of color developer was added, mixed well, and centrifuged. The supernatant was collected, and its absorbance was measured at 540nm. Addition-ally, 0.5 ml of the extracted enzyme solution was diluted to 100 ml with distilled water. Then, 2 ml of the diluted solution was used to measure the soluble protein content at 595nm using Coomassie Brilliant Blue G-250.

 Comments 5. “the net photosynthetic rate (Pn), transpiration rate (Tr), intercellular CO2 concentration (Ci), and stomatal conductance (Gs) of sweet potato leaves were measured using Li-6400 photosynthetic instrument from 9:00 a.m. to 11:00 a.m.” Specify why this range.

Response 5: Thanks for your good questions. The best time to measure photosynthesis should be the time period when plant photosynthesis can reach the highest efficiency under the conditions of appropriate light intensity, light cycle, temperature, moisture and carbon dioxide concentration. The light intensity in the morning is weak; the light intensity at noon is too strong; the light intensity decreases in the evening, the photosynthesis rate decreases, and the plant turns to respiration to maintain life activities. The photosynthetic instrument test needs to be carried out after the induction period of photosynthesis ends and photosynthesis reaches a steady state, otherwise it will lead to inaccurate instrument measurement results. Therefore, we uniformly measure when photosynthesis is stable from 9:00am to 11:00am.

Comments 6. “The soluble sugar and starch content were determined using the anthrone colorimetric method; the protein content was determined using the Kjeldahl N method.” please specify the methods.

Response 6: Thanks for your good suggestions. Based on your suggestions, we have made the following additions to the determination methods section.

Line 145-154: Firstly, 0.5 g of sample powder was weighed and added to 80% ethanol. The mixture was extracted 2-3 times at 80 °C, and the supernatant obtained was used to measure soluble sugars. The remaining residue was then extracted with distilled water and HClO4 in a boiling water bath, and the supernatant was used as a starch test solution. A 2 ml test solution was mixed with anthrone-H2SO4 reagent and heated in a boiling water bath for 10 minutes. After cooling, the absorbance was measured at 620nm [28]; In addition, 0.2 g of powder sample was placed in a digestion tube. 10 ml of concentrated H2SO4 was added to digest at 420 °C for 1.5 h, and then H2O2 was added for re-action. After cooling, the volume was 50 ml with distilled water. The solution was subsequently processed using a Kjeldahl N autoanalyzer, distilled, and titrated, and the titration data was recorded to calculate the protein content [26].

Comments 7. “Microsoft Excel 2016 and SPSS (version 22.0) were used to analyze the experimental data, and conduct variance analysis.” specify the statistics used, anova? please this is very important.

Response 7: We greatly appreciate your valuable suggestions. According to your suggestions, we have made the following supplements and explanations in the statistical analysis section.

Line 164-168: Microsoft office excel 2016 is used for data entry, organization and preliminary calculation. The data were analyzed by two-way ANOVA and Duncan's multiple range test (P < 0.05) using SPSS 22.0 to evaluate the effects of N and K treatments and their interactions on various physiological parameters of sweet potato plants. GraphPad Prism9.5 drawing.

Comments 8. “Conclusion”, add some key value of your results.

Response 8: Thank you for raising important questions about the manuscript. We have made revisions to the Conclusion section to address your concerns.

Line 463-480: This study found that a certain amount of K fertilizer could effectively increase the activity of GS, promote N metabolism, and then improve the utilization rate of N fertilizer when the N application rate was 120 kg·ha-1. The application of appropriate amounts of N and K fertilizers can improve the photosynthetic characteristics of sweet potato leaves and increase the accumulation of photosynthetic products. These photo-synthetic products are transported to the root tubers through the stems, promoting the expansion of the root tubers and the accumulation of nutrients, improving the yield and quality of the root tubers. For farmers, the appropriate combined application of N and K is an effective strategy to optimize sweet potato growth and maximize yield. This fertilization method not only improves the utilization efficiency of N fertilizer but also improves the photosynthetic capacity of the plant, thereby increasing the yield and quality of tubers. However, excessive N application can lead to excessive leaf area index, reduced photosynthetic efficiency, and excessive vegetative growth, which has a negative impact on tuber development. K fertilizer is key in balancing vegetative growth and ensuring optimal yield and quality. Therefore, in actual implementation, farmers should focus on the balance and moderate application of N and K fertilizers to achieve the best results in sweet potato cultivation.

Round 2

Reviewer 3 Report

Comments and Suggestions for Authors

The authors improved the manuscript, i suggest the publication in Agronomy

Author Response

Dear Reviewer,
Thank you for your positive feedback and recommendation for publication in Agronomy. We greatly appreciate your time and effort in reviewing our manuscript. Your suggestions and insights have been invaluable in improving the quality of our work.

Sincerely,
SX